# Dare to Ask! A Model for Teaching Nursing Students about Identifying and Responding to Violence against Women and Domestic Violence

Leah Okenwa Emegwa [1,*], Stéphanie Paillard-Borg [1], Inger Wallin Lundell [1], Anna Stålberg [1], Maria Åling [1], Gabriella Ahlenius [1] and Henrik Eriksson [1,2]

1   Department of Health Sciences, Swedish Red Cross University, P.O. Box 1059, SE-141 21 Huddinge, Sweden
2   Section for Health Promotion and Care Sciences, University West, SE-461 32 Trollhättan, Sweden
*   Correspondence: leok@rkh.se

**Abstract:** The role of nurses in identifying and responding to family violence and violence against women has long been established. However, nurses' readiness to fully assume this role remains low due to various barriers and the sensitive nature of the subject. As part of capacity building to address this problem, an additional national qualitative learning target, i.e., to "show knowledge about men's violence against women and violence in close relationships", was introduced into the Swedish Higher Education Ordinance for nursing and seven other educational programs between 2017 and 2018. The aim of this paper is to describe how the national qualitative learning target is incorporated into the undergraduate nursing curriculum at the Swedish Red Cross University College. An overview of relevant teaching and learning activities and how they are organized is first presented, followed by the presentation of a proposed didactic model: Dare to Ask and Act! The model details a step-by-step progression from facts and figures, including the role of gender norms, to recognizing signs of abuse in complex clinical situations, as well as developing skills that enhance the courage to ask and act. Due to the sensitive nature of violence victimization, the proposed model reflects the importance of making the subject a reoccurring theme in undergraduate nursing education in order to boost nursing students' interests and confidence to "Dare to Ask and Act!". The model also shows that making the subject a recurring theme can be achieved with minimal disruptions to and without overcrowding an existing curriculum.

**Keywords:** nursing education; violence; curriculum; courses; alignment; progression





## 1. Background

Nurses are important stakeholders for identifying and responding to violence against women (VAW) and domestic violence (DV) [1–3]. There are currently increasing calls to integrate VAW/DV screening as part of standard nursing care practice and include it in nursing curriculums [4,5]. Although the terms VAW and DV are used interchangeably, there are certain differences related to scope. VAW includes "any act of gender-based violence that results in or is likely to result in physical, sexual or psychological harm or suffering, as well as threats of such acts, coercion or arbitrary deprivation of liberty, whether in public life or in private life" [6,7]. Other forms of VAW include honor violence, trafficking, and forced prostitution, as well as diverse harm in traditional practices around the world, such as female genital mutilation, breast ironing, dowry-related violence, and widowhood practices [8,9]. DV, on the other hand, is described as any controlling and/or violent act (physical, sexual, emotional, financial, or neglect) inflicted by a person on another who forms part of a household unit or a close relationship [10,11]. DV is often a pattern of actions ranging from subtle acts to serious crimes, such as anything from being ridiculed to being subjected to rape or serious threats [12]. DV includes child abuse, abuse of the elderly, and violence between partners (former or current), regardless of whether the perpetrator

shares or has shared a home with the victim or not [6,13]. DV against a woman is, thus, a form of the broader VAW. VAW and DV are complex problems [13] driven by gender norms and expectations [14,15]. VAW and DV are, thus, easily neglected and misunderstood in, for example, same-sex relationships due to factors like the masculine-feminine dichotomy and established stereotypes of males as perpetrators and females as victims [16–19].

The consequences of VAW/DV cut across physical, reproductive, and psychosocial health and wellbeing. Research shows that compared to others, people at risk of or exposed to VAW and DV tend to seek healthcare more often due to injuries, complications arising from the violence, and also in the hopes of finding help; unfortunately, however, violence as an underlying problem is often never identified [20,21]. According to Watts and Mayhew (2004), three categories of victims are likely to be encountered in healthcare, i.e., those who disclose abuse or the fear of likely abuse, those who do not disclose abuse but show abuse-related signs and symptoms (for example, bruises, tears, scrapes and reproductive health complications including low birth weight, repeated miscarriages, etc. [22]. The final category includes abuse victims who neither disclose nor exhibit signs or symptoms of abuse [22]. Barriers to disclosure include, among other factors, concerns about the safety of the individuals themselves or their children and fear of reprisals from the perpetrator [23].

The role of nurses in VAW/DV screening and response has long been established. For example, given the core principles of nursing practice, nurses often have close contact with patients [24,25]. Research also shows that nurses are often patients' first and/or only point of contact, depending on the context [4,24,26]. Unfortunately, nurses encounter barriers that mitigate against their strategic role in identifying cases of VAW and DV; examples include a lack of time, lack of training, and lack of clear guidelines and procedures [27]. Other barriers are personal beliefs [25,28–30], the perception that screening for partner violence is an invasion of privacy, and the belief that people exposed to VAW and DV are likely to return to the perpetrator [31]. Moreover, although guidelines and standards exist in healthcare and often provide a good framework for responding to identified cases, they do not necessarily provide the basic knowledge and competence required for VAW/DV screening and response [32].

As part of Sweden's national strategy to promote gender equality and address VAW and DV [33], the Swedish Higher Education Ordinance was adjusted in 2017 to include an additional degree objective on the subject for eight educational programs [34–37]. The programs are for professionals who regularly encounter victims and perpetrators of violence and include law, social workers, physiotherapy, psychology, dentistry, dental hygiene, medicine, and nursing. The additional degree objective broadly formulated to "demonstrate knowledge of men's violence against women and violence in close relationships" resulted in the mandatory teaching of VAW/DV in Swedish nursing programs [38,39]. Although the additional degree objective is considered broad, it allows room for educators to design content relevant to the scope and practice of each individual profession [40]. Furthermore, the wording of the additional degree objective shows the distinction between VAW generally and the violence occurring among people with close relationships., i.e., DV.

Given that the quality assurance of Swedish higher education is conducted via regular reviews and assessments, the additional national degree objective resulted inthe need for affected educational programs to ensure the presence and visibility of VAW/DV content in their course plans.-in, [33]. Guidelines and qualitative targets for education are regulated by the Swedish Higher Education Act and the Higher Education Ordinance, and the Swedish Higher Education Authority (Universitetskanslersämbetet, UKÄ) is responsible for all quality assurance [33]. While the inclusion of VAW and DV in nursing curricula will no doubt foster capacity building, a common challenge in nursing education is the risk of content saturation and an overcrowded curriculum as educators race to also keep up with emerging aspects of health and healthcare [41,42]. This paper aims to describe the implementation process of the degree objective on VAW/DV in nursing education at the Swedish Red Cross University and some of the lessons learned, including the development of a model for teaching the subject in the university's nursing program.

## 2. Method

This study is based on qualitative data from the curriculum, course syllabi, and course activities concerning VAW/DV from semesters one through six of a three-year undergraduate nursing program.

### 2.1. Context and Materials

The Swedish Red Cross University (SRCU) has been training nurses since 1867 and currently offers the following courses: single courses in nursing, a three-year undergraduate education, and master's degrees in three specialist nursing programs, i.e., intensive care nursing, psychiatric nursing, and infectious disease nursing. The undergraduate program is a three-year program (i.e., six semesters) and totals 180 in the ECTS (European Credit Transfer and Accumulation System). Each semester, students take a 30 ECTS course divided into modules. The undergraduate nursing program complies with the European Union's (EU's) requirements for nursing education as stated in the European Economic Area EEA Agreement (and later European Economic Community directives), i.e., to enable service within EU countries. A detailed description of the curriculum and courses is available on the university's website [43].

Teaching and learning activities on VAW/DV were already a part of the nursing education at SRCU before 2017. However, SRCU's need to meet up with future VAW/DV-related quality assurance inspections meant that an appraisal of existing teaching and learning activities was necessary in order to properly implement the additional objective on VAW/DV. Two major challenges were in focus; firstly, VAW/DV are broad subjects, while the second is the need to implement the national directive without overcrowding the curriculum [41,42]. Therefore, rather than wait to design a totally new course for VAW/DV, a collegial decision was made to first review the existing curriculum and course syllabi and to conduct the mapping of existing VAW/DV teaching and learning activities. The rationale behind this decision was that since nursing training is already designed to prepare students to recognize and respond to actual or potential health problems [44], there were bound to be parts of the curriculum in which the VAW/DV content could be considered a "natural fit". The outcome of the review and mapping would, thus, be used to design the content and structure of VAW/DV teaching and learning activities.

### 2.2. Process

The process had two phases. The goal of the first phase was to identify areas in the undergraduate nursing program where VAW/DV content is a natural fit; thus, the curriculum and all individual course syllabi from semester one through to semester six were read through. Relevant texts in the course descriptions where VAW/DV could fit were extracted, collated, and arranged in order, i.e., from the first semester to the sixth semester. The text extracts were then analyzed using content analysis, and the data analysis process is described in the appropriate section below. In the second phase, a mapping of VAW/DV teaching and learning activities was conducted via a shared document. The authors, all of whom were engaged in teaching VAW/DV in one form or the other, documented information regarding the VAW/DV content, teaching resources, textbooks and recommended reading, examination forms, the total number of hours for each activity, and the lecturer (i.e., whether external or internal in order check the magnitude of teacher resources needed). All relevant information was gathered for years one to three, i.e., from the first semester to the sixth semester.

*2.3. Data Analysis*

The texts that were extracted in the first phase served as meaning units for content analysis. Content analysis, defined as "any technique for making inferences by objectively and systematically identifying specified characteristics of messages" [45], is found useful when the objective is to examine patterns in documents [46,47]. Content analysis is useful when there is a need to provide a description of the characteristics of the content of a document [48]. For this study, content analysis inspired by Lundman and Graneheim's five-step method [49] was conducted; the steps include identifying the meaning units, condensation, generating codes, and grouping codes in order to create categories and then sub-categories [49]. The meaning units (i.e., texts extracted from curriculum and course syllabi) were condensed to identify the manifest content (i.e., close to text content) and latent content (i.e., interpretation of the underlying meaning). Given that the task was to identify areas in the syllabi where the VAW/DV content could fit, the final focus was on the latent content. According to Graneheim and Lundman (2004), while categories are used to represent the manifest content, themes are used to present latent content [49]. The condensed meaning units (i.e., text from the curriculum and course syllabi) were, thus, thereafter, used to create sub-themes and an overall theme based on collegial discussions and reflection; see Table 1.

**Table 1.** Content analysis of curriculum and course syllabi showing meaning units, codes, sub-themes and themes.

| Year | Semester | Data Extract from a Summary of the Course Description | Condensed Unit Description Close to the Text | Codes | Sub-Theme | Theme |
|---|---|---|---|---|---|---|
| 1 | I | ...relate nursing care to societal challenges and their consequences for health and healthcare such as vulnerability, inequalities, including gender inequality. | i. Societal drivers of inequalities and consequences on health and healthcare | Population health challenges | -Facts, attitudes, values and policies | Advancement in knowledge and readiness to act |
| | II | "...Nursing concepts and models, ...nursing care process, person centered care as well as international and national public health goals and policy documents." | Nursing care processes and models related to public health goals and policies | Public health policies, and nursing care. | | |
| 2 | I | Identify individual-level mechanism of ill-health locally and globally, and to conduct nursing care activities that promote health and prevent ill-health. | Global and local perspectives of causes of ill-health and nursing care activities to address them | Nursing care and health of individuals globally, locally. | Training general skills and abilities | |
| | II | ...students practice skills for conducting nurse-patient health talks related to lifestyle, and also how to inquire about exposure to VAW/DV. | Clinical skills for Nurse-patient interaction/talks and screening for exposure to VAW/DV. | Skills for nurse-patient talks, including screening for violence. | | |
| 3 | I | ...develop ability to evaluate and to have the right nursing care approach. Perspectives on health equality and equity as well as sustainable development are included in the course. | Nursing skills for equality, equity, observation, decision making and evaluation. | Interpret patient circumstances and apply nursing care skills. | Perceive signs in complex situations | |
| | II | Student should, via supervision, plan, conduct and report an independent research work in nursing care, to be presented and evaluated during a seminar. | Independent research thesis work in nursing care | Possibility to choose VAW/DV for degree thesis | | |

## 3. Results

One overarching theme and three subthemes emerged, i.e., advancement in knowledge and a readiness to act, with each sub-theme representing the study focus of each year of the three-year program, i.e., facts, policy, attitudes, values; training general skills and abilities; and perceive signs in complex situations (see Table 1). The theme and sub-themes, thus, reveal a structure progressively advances students' knowledge. Attention was then turned to the findings from the mapping of teaching and learning activities conducted in the second phase of the process.

The mapping revealed that VAW/DV was already successfully embedded and delivered within each of the existing thirty ECTS courses per semester from the first semester through to the third semester. The VAW/DV contents were in alignment with course descriptions and were delivered in a way that advanced theoretical knowledge without taking up much space or time, i.e., about three to five hours per semester, see Table 2. For example, in line with the course theme for year one, VAW/contents are introduced in the first-semester course as part of inequalities and population health (public health) problems (see Tables 1 and 2). Furthermore, international and national policy documents on VAW/DV are taught as part of documents that have influenced health policy, planning, and systems over the years, which is taught in the second semester. However, for semesters four and five, findings from the mapping showed that there was a need to adjust existing theoretical content and add practical and hands-on content (Table 2).

*Application of the Findings*

The identified theme and sub-themes served as a guide to design relevant VAW/DV contents, adjusting existing contents where necessary to ensure knowledge progression. To apply these findings, the authors were guided by three key principles; the first principle was the importance of identifying VAW/DV contents relevant for nursing care, which was conducted by revisiting original text extracts from the curriculum and reviewing the literature [40]. The second principle was to add identified relevant VAW/DV contents in a manner that would neither overcrowd nor disrupt the current curriculum or lead to significant changes [42]. The third principle was to take into cognizance research findings that suggest the need to build students' confidence to ask intimate questions by keeping the subject in focus throughout the pre-licensing period [50–53]. The authors' inspiration to bear the third principle in mind was further reinforced by the observation that relevant VAW/DV content has already been successfully embedded into courses in the first, second, and third semesters. More practical and hands-on content was added in semesters four and five. Practical moments were provided in the form of seminars and workshops based on various simulations (short films), case studies, as well as experiences and observations during their clinical placements. Other practical hands-on aspects introduced in semesters 4 and 5 included but were not limited to intersectoral collaboration, guidelines for making a report of concern, and how to make such reports, especially for vulnerable groups such as children. In the sixth and final semesters, there are opportunities for interested students to write their degree thesis on VAW and DV.

Table 2 shows current teaching and learning activities. VAW/DV contents make up between three to five hours per semester in the first one and a half years of the program. The number of teaching hours increases to about seven to eight hours per term in the later one and half a year of the program when students start practicing nursing skills for VAW/DV screening, response, and intersectoral collaboration. To visualize how teaching and learning activities are organized to align with the curriculum and advance knowledge, a diagrammatic representation was created and is presented as the "Dare to Ask and Act" didactic model in Figure 1; the themes for each year and semester are presented on the *Y*-axis and the specific VAW/DV contents for each semester are presented on the *X*-axis.

**Table 2.** Teaching and learning activities related to violence against women (VAW) and domestic violence (DV).

| Year | Year Theme in Relation to VAW/DV | Semester | Teaching Content Related to VAW/DV | Teaching and Learning Activities | Number of Hours * | Examination |
|---|---|---|---|---|---|---|
| 1 | Facts, attitudes, values, and policies | 1 | Introduction to the subject of VAW and DV, gender perspectives. | Lectures | 2 | Group assignment |
| | | | | Online course from NCK [1] | 3 | Mandatory submission of certificate of completion |
| | | 2 | Human rights, laws, Iinternational and national population health goals and policy documents. Specific vulnerable groups like LGBTQ [2], older persons etc. | Lectures | 2 | International and national documents on VAW/DV included in course examination. |
| | | | | Online course on violence against older persons from NBHW [3] | 1 | Mandatory submission of certificate of completion |
| 2 | Training general skills and abilities | 3 | Global perspectives on violence against women and domestic violence including forms, norms and practices; VAW/DV in humanitarian situation. | Lectures | 4 | There are specific questions on VAW/DV in the written individual examination. |
| | | 4 | The role of healthcare, routines and guidelines, including screening and reporting | Lectures, workshop, short films, and reflections related to screening. | 5 | The final individual written examination includes questions on VAW/DV. |
| | | | | Online course-Barnfrid [4] | 3 | |
| 3 | Perceive signs in complex situations | 5 | Routines, guidelines, screening, and reporting (continued); intersectoral collaboration | Workshop on clinical assessment and nursing care measures for VAW/DV, and intersectoral collaboration with relevant stakeholders. | 3 | Written take-home examination in which one of the questions is a case study for which students must: -assessing VAW/DV, including child abuse/neglect -write a referral to relevant stakeholders, and -reflect on the role of the health sector from an ethical standpoint. *Note: A pass in the seminar is a prerequisite for passing the clinical placement in primary care.* |
| | | | | Reporting suspected abuse/child neglect is included in child health care using three case studies. | 3 | |
| | | | | Seminar: Identifying VAW/DV and intersectoral collaboration is included in this seminar centered on clinical placement in primary care. | 1 | |
| | | 6 | Degree thesis | Possibility to research VAW/DV Via degree thesis for interested students [5] | | |

* The number of hours spent by students for preparing for diverse teaching and learning activities is not included in the total number of hours presented. [1] NCK—National center for knowledge on men's violence against women. [2] LGBTQ—Lesbian, gay, bisexual, transgender, and queer. [3] NBHW—National board of health and welfare (Socialstyrelsen). [4] Barnafrid—National center for knowledge concerning violence against children. [5] Researchers in the field available to supervise students interested in writing their thesis on the subject.

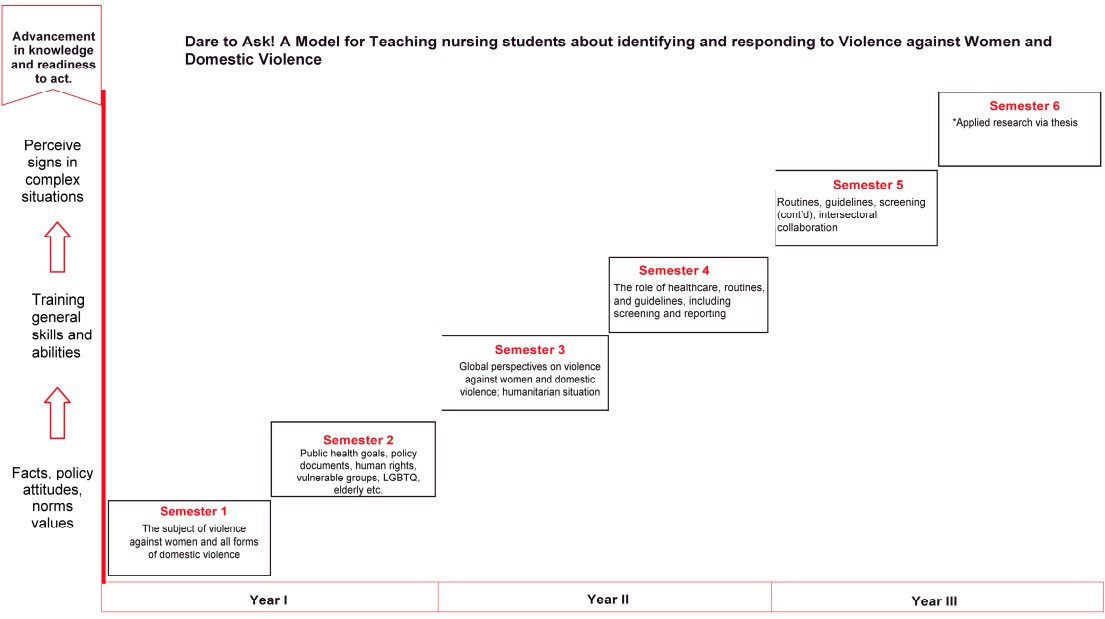

**Figure 1.** Dare to Ask and Act model to teach about violence against women and violence in close relationships.

## 4. Discussion

VAW/DV contents were incorporated within the framework of existing teaching and learning activities and with minimal disruption to the curriculum. This was possible because nursing training is already designed to prepare students for the role of promoting health, preventing diseases, and helping individuals or groups assess their responses to actual or potential health problems [44]. Moreover, nurses encounter individuals at risk of or already exposed to VAW/DV in various nursing care contexts. Findings from the content analysis of the documents and mapping of teaching and learning activities showed that there are many areas in the existing curriculum where VAW/DV is a natural fit. According to Wyatt, McClelland, and Spangaro (2019), it is possible to embed the VAW/DV content within diverse scenarios in nursing education [50]. Teaching VAW/DV should, therefore, not be seen as separate content but one that can occur within the framework of existing teaching and learning activities.

As already stated, e several areas in the curriculum where VAW/DV could fit without the risk of curriculum overcrowding or displacing other equally important contents were identified. By introducing VAW/DV into different aspects of nursing education, the subject becomes recurrent in different forms as students' education progresses from theoretical knowledge to acquiring necessary clinical skills. By gradually introducing the subject over the course of the nursing program, building blocks that progress from facts to clinical skills for screening and response are formed. Furthermore, students receive the opportunity to encounter VAW/DV content several times throughout the period of the three-year education. VAW/DV, as a recurrent theme in nursing education, may help to address a major barrier to screening, i.e., nurses' discomfort with screening due to fear of offending or violating the patient's privacy [31,32]. By providing an opportunity to encounter VAW/DV in different aspects of nursing training, the subject remains fresh, and students obtain a broader understanding of all types of violence rather than a limited perspective commonly seen in many nursing curricula [54]. Nursing students need more than just stand-alone lectures, as these do not lead to the development of the required knowledge and necessary clinical skills [51]. Studies show that to help students develop confidence in handling difficult and intimate nursing care aspects, they should be given the opportunity to severally encounter these subjects during their education [50,55].

Despite evidence that nurses are more likely to ask about VAW/DV if they know where to refer the victim [56,57], one shortcoming in nursing education is the absence of content on intersectoral collaboration [54]. A Swedish study of nurses showed that nurses often responded to VAW/DV by sending the patient to the doctor; it was not clear if this practice was due to nurses' lack of knowledge, a belief that doctors have the answers, or simply a way to shift responsibility [58]. The need to have clear routines and guidelines for how the care team manages patients' disclosure so that patients are not made to retell their story, as has been emphasized by the National Center for Knowledge on Men's Violence Against Women, NCK [59] In the Dare to Ask and Act model (see Figure 1 and Table 2), the fourth and fifth semesters are dedicated to screening, response, and intersectoral collaboration, such as with the police, social welfare services, and women's shelters, among others. Students also learn about the principle of autonomy in decision making for adults and, especially for children, and the process of writing a report of concern to the relevant authorities. The overall design and content of VAW/DV teaching and learning activities must be a well-thought-out and in-depth education that leads to measurable theoretical knowledge, relevant nursing skills, and the courage to act.

Some of the challenges related to including VAW/DV in nursing education are worth mentioning, for example, staffing and teaching materials [40]. Although textbooks that address VAW/DV from a nursing care perspective are not many, it is possible to combine the few available ones with teaching materials from other digital resources, such as information materials, short online courses, and short films available on the websites of many stakeholder organizations. Some examples used in this case are textbooks on nursing care principles, home-care nursing [60], emergency care [61], and public health [62]. Because of the interpersonal nature of the subject and the need to ensure patient safety, there is a limited chance for students to meet real patients. There is, thus, a need to use variations in teaching forms such as simulations, films, and other materials specifically made from a nursing care perspective. For example, films available on the website of the National Center for Knowledge on Men's Violence Against Women, NCK [59] were used. The NCK has produced several short films showing interactions between patients exposed to VAW/DV and nurses in different nursing care contexts. The patients in the films have diverse sociodemographic backgrounds, including the elderly, immigrant groups, and women of reproductive age, among others. Similarly, films and materials from Barnafrid, i.e., the National Centre for Knowledge Concerning Violence Against Children [63], Ecpat, and the Ombudsman for Children, were used as teaching materials. The films show children's vulnerability, the importance of listening to them, and the need to identify children at risk. The National Board of Health and Welfare (Socialstyrelsen) has a short digital course on violence against the elderly. All films are often followed by time for reflection, observations, etc., via a workshop or seminar. The films provide a good starting point for students and teachers to discuss ethical aspects, such as the patient's autonomy in decision-making regarding whether or not to contact authorities or other stakeholder organizations. An examination of student knowledge varies from seminars to group assignments and are specific questions in individual written examinations; see Table 2.

Research results must be rigorous and trustworthy; thus, the responsibility to ensure reliability lies on the researchers [64]. The authors took several steps to ensure reliability. For example, all authors were involved in identifying relevant texts in the courses which they were responsible for. The first author was responsible for coding the text extracts, and all authors were involved in several discussions and reflections. Thereafter, the last author cross-checked the codes, and adjustments were made where necessary. In the final stages, all the authors agreed regarding the final theme and sub-themes; these are the basis on which the proposed model was developed. It is, thus, hoped that in addition to providing a structure for advancement in knowledge and readiness to act, the "Dare to Ask and Act" model for teaching nursing students about VAW/DV (Figure 1) can also serve as a basis for collegial reflection and dialogue about how teaching should develop. Some crucial questions for such collegial dialogue and reflection are, for example, what has been

missed? What aspects can further be developed? Are there shortcomings or possibly any missing relevant content in the current model, and how can they be addressed? Recent reports show that some of the challenges of implementing the additional degree objective reported by nursing educators in Sweden include ambivalence to the subject among some faculty members, a perception that the degree objective is too broad, and a continued tendency to use VAW and DV interchangeably, thus resulting in disagreements regarding its scope among the faculty [40,65,66]. Hopefully, a continuous evaluation of the current approach will ensure continuous dialogue among faculty members. It is also hoped that continued faculty dialogue about the proposed model can serve as quality assurance of the model and lead to improvements so that nursing students are equipped to acquire relevant knowledge and skills to "Dare to Ask and Act". In doing so, no patient exposed to or at risk of VAW/DV will be left without help.

Finally, considering that nursing education is anchored on theory and clinical exposure, the consistent reports about low readiness to screen for VAW/DV and inadequate responses to VAW/DV within healthcare, including Swedish healthcare [67–69], raise concerns about potential knowledge-to-practice gaps for nursing students. Although Swedish health facilities are encouraged to have written guidelines and action plans (mandatory for some), some healthcare professionals report a lack of or non-existence of clear routines and guidelines as a barrier to screening and response [58,67,68,70]. Considering that newly trained nurses often quickly become the products of their work environment [40], it is hoped that the use of the "Dare to Ask and Act" model in nursing education may inspire health facilities that collaborate with nursing schools for students' clinical placements to establish clearer VAW/DV routines and guidelines and increase their readiness to screen.

**Author Contributions:** Conceptualization, L.O.E. methodology, L.O.E. and H.E.; formal analysis, L.O.E. and H.E.; investigation, L.O.E., S.P.-B., I.W.L., A.S., M.Å., G.A. and H.E.; resources, L.O.E., S.P.-B., I.W.L., A.S., M.Å., G.A. and H.E.; data curation, L.O.E., S.P.-B., I.W.L., A.S., M.Å., G.A. and H.E.; writing—original draft preparation, L.O.E., S.P.-B., I.W.L., A.S., M.Å., G.A. and H.E.; writing—review and editing, L.O.E.; visualization, L.O.E. and H.E.; supervision, L.O.E. and H.E.; project administration, L.O.E. All authors have read and agreed to the published version of the manuscript.

**Funding:** This research received no external funding.

**Institutional Review Board Statement:** Not applicable.

**Informed Consent Statement:** Not applicable.

**Data Availability Statement:** Data materials used for this study are curriculum and course plans available at rkh.se.

**Public Involvement Statement:** No public involvement in any aspect of this research.

**Guidelines and Standards Statement:** This manuscript was drafted standard reporting guidelines for qualitative research.

**Conflicts of Interest:** The authors declare no conflict of interest.

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
