# Peer review of "Dare to Ask! A Model for Teaching Nursing Students about Identifying and Responding to Violence against Women and Domestic Violence"

_nursrep, doi:10.3390/nursrep14010046_

Round 1
Reviewer 1 Report
Comments and Suggestions for Authors
The manuscript describes the re-organization (expansion?) of the nursing curriculum in one Swedish college to include the topic of violence against women (VAW) and domestic violence (DV). However, the manuscript in its present form generates the impression that the authors’ primary aim was to show their compliance with a national requirement rather than share their experience and summarize salient points for others involved in similar tasks.
Major remarks
The unfortunate title should be recomposed as absolutely nobody should teach violence against women („Model for Teaching Violence against Women…).
The first problem is that the two large but only partially overlapping topics are treated by the authors together without any description as to what specific issues were prioritized in the curriculum. Violence against women (VAW) includes various cultural practices such as female genital mutilation, breast ironing, dowry-related violence, and other traditional practices around the world; trafficking and forced prostitution of females in and out of domestic settings; abuse of women of all ages including sexual abuse, rape in all sorts of circumstances such as date rape, marital rape, mass rape of females in war, etc. to name a few as defined by the UN in 1993. Domestic violence (DV) refers to a more restricted set of violent acts against members of households but this includes not only women but children and elderly (and males) as well. DV is a more loose term not having an international guideline about its definition. The WHO issued a plan of action in 2016 to address interpersonal violence which includes not only women but children as well. The authors should clearly specify what topics and issues are included in the curriculum; for example, will nurses be able to recognize and address female genital mutilation (ob/gyn nurses may encounter cases even in Sweden)? marital rape or mass rape suffered by refugee women? Does the curriculum cover child abuse in general or only female child abuse?
The authors need to explain their aim when expanding the curriculum beyond the national degree objective because the national objective clearly calls for „demonstrate knowledge …” from which it does not follow that „skills and ability to perceive signals in complex situations” (L169) should be taught or that teaching should aim at preparing for „readiness to act” (Fig.1). Knowledge, attitudes and skills comprise competences, and – as much as the authors revealed it – building competences for nurses was NOT prescribed in the national guideline.
Related to the point above, another problem is that the authors do not demonstrate how the implemented changes are related to what had been taught BEFORE (the authors state this in L102-104) the national degree objective was introduced. The authors should show the comparison – preferably in a table format – between what had been taught before (semester, teaching content, topics, number of hours) and the newly expanded version. They should also introduce the principles by which they decided 1) what topics to add; 2) why the total number of teaching hours should be 27 hours (not 25 or 30 or some other number); 3) why the topic must be taught in ALL semesters (Table1) given that overcrowding the curriculum was an issue.
Another question: how the new nursing competences are related to competences of medical doctors treating the same patients in the same settings? What are the competences of nurses vs doctors regarding actions in suspected cases of violence? How is INTRAsectoral collaboration (within healthcare) taught? Can it happen that a nurse suspects WAW/DV, dares to ask questions from the patient and reports it to social care or other services without consulting the patient’s doctor?
Missing from the manuscript is the information regarding the level of higher education at which curricular changes took place: bachelor or master? If bachelor as it can be assumed from Tables 1 and 2, what is taught at master level?
Minor remarks:
Affiliations should be corrected as the names of institutes are capitalized in English.
What is „neglect of barn”? Table 2 Semester 5
English words should be used (Semester instead of Termin, Fig.1)
Fig. 1. does not require colors.
Table 2 contains Swedish text.
Comments on the Quality of English LanguageIncluded in the review.
Author Response
Comments and Suggestions for Authors
The manuscript describes the re-organization (expansion?) of the nursing curriculum in one Swedish college to include the topic of violence against women (VAW) and domestic violence (DV). However, the manuscript in its present form generates the impression that the authors’ primary aim was to show their compliance with a national requirement rather than share their experience and summarize salient points for others involved in similar tasks.
Major remarks
Comment: The unfortunate title should be recomposed as absolutely nobody should teach violence against women (Model for Teaching Violence against Women…).
Reponse: Two titles are suggested as follows
- Dare to Ask! A Model for Teaching screening and intervention for Violence against Women and Domestic Violence in Nursing Education
- Dare to Ask! A Model for Teaching nursing students about identifying and responding to Violence against Women and Domestic Violence
Comments: The first problem is that the two large but only partially overlapping topics are treated by the authors together without any description as to what specific issues were prioritized in the curriculum. Violence against women (VAW) includes various cultural practices such as female genital mutilation, breast ironing, dowry-related violence, and other traditional practices around the world; trafficking and forced prostitution of females in and out of domestic settings; abuse of women of all ages including sexual abuse, rape in all sorts of circumstances such as date rape, marital rape, mass rape of females in war, etc. to name a few as defined by the UN in 1993. Domestic violence (DV) refers to a more restricted set of violent acts against members of households but this includes not only women but children and elderly (and males) as well. DV is a more loose term not having an international guideline about its definition. The WHO issued a plan of action in 2016 to address interpersonal violence which includes not only women but children as well. The authors should clearly specify what topics and issues are included in the curriculum; for example, will nurses be able to recognize and address female genital mutilation (ob/gyn nurses may encounter cases even in Sweden)? marital rape or mass rape suffered by refugee women? Does the curriculum cover child abuse in general or only female child abuse?
Response:
- Thanks for the comments, we have now added a text with some concrete examples, see lines 37-40.
- The text in the previous submission addresses most of the other issues raised, so yes, nurses will be able to recognise various forms of VAW and DV. For example, in the original submission, we provided clear distinction between VAW and DV in liness 44 to 47 and risks posed to children and elderly.
- To indicate that vulnerability cuts across gender, the expression “…inflicted by a person to another who form part of a household unit or a close relationship…”, uses “person” rather than “women”, see lines 40 - 42
- In lines 47-51 we address DV in same-sex relationship.
- In Table 2, year 2 semester 3, the content is summarized as “Global perspectives on violence against women and domestic violence including forms, norms and practices; VAW/DV in humanitarian situation.
Comment:
The authors need to explain their aim when expanding the curriculum beyond the national degree objective because the national objective clearly calls for „demonstrate knowledge …” from which it does not follow that „skills and ability to perceive signals in complex situations” (L169) should be taught or that teaching should aim at preparing for „readiness to act” (Fig.1). Knowledge, attitudes and skills comprise competences, and – as much as the authors revealed it – building competences for nurses was NOT prescribed in the national guideline.
Response:
- An additional text has been added to explain that the additional degree objective was broadly formulated to allow room for affected educational programmes to design content relevant to the scope and practice of each individual profession, see lines 84-86.
- While the additional degree objective says ”demonstrate knowledge”, for a nurse ”knowledge” would mean not only theoretical knowledge, but also clinical skills to provide nursing care irrespective of the situation, circumstances or context in which the patient is encountered.
- Skills and the ability to perceive signals in complex situations are thus a combined outcome of the content analysis of the nursing curriculum document at The Swedish Red Cross University which shows the knowledge progression. It will be utterly impossible for nurses to play their role in identifying and responding to VAW/DV without acquiring the necessary skills and comptences beyond theoretical knowledge.
Comments:
Related to the point above, another problem is that the authors do not demonstrate how the implemented changes are related to what had been taught BEFORE (the authors state this in L102-104) the national degree objective was introduced. The authors should show the comparison – preferably in a table format – between what had been taught before (semester, teaching content, topics, number of hours) and the newly expanded version. They should also introduce the principles by which they decided 1) what topics to add; 2) why the total number of teaching hours should be 27 hours (not 25 or 30 or some other number); 3) why the topic must be taught in ALL semesters (Table1) given that overcrowding the curriculum was an issue.
Response:
- Although a table showing what had been taught before and after the national degree objective was introduced may be interesting for the sake of comparism, the authors do not consider a comparism as the focus of this article. Our intention is to show how teaching has been organised as part of implementing a new national directive. Furthermore, We have two tables with contents that are relevant to the main focus of the article.
- The guiding principles and the rationale behind content to be included are found in lines 127 – 131, and 194 – 201
- The total number of hours is the observed total in the entire program following mapping and not a predetremined number. The authors were interested in knowing the total numbers of hours devoted to VAW/DV content for the purpose of not displacing other contents, and also for alignment, especially to ensure that the total workload for students matches the total course credit. We found that by keeping the subject reoccuring within the context of various courses throughout the three year programme, students got exposed to significant level of VAW/DV content, including hands-on application, without (a)having to remove other contents in the curriculum to make room for VAW/DV; (b) curiculum overcrowding.
Comment:
Another question: how the new nursing competences are related to competences of medical doctors treating the same patients in the same settings? What are the competences of nurses vs doctors regarding actions in suspected cases of violence? How is INTRAsectoral collaboration (within healthcare) taught? Can it happen that a nurse suspects WAW/DV, dares to ask questions from the patient and reports it to social care or other services without consulting the patient’s doctor?
Response:
The medical programme (and thus doctors) and other relevant programmes within healthcare are also included in the implementation of the degree object, see lines 78 – 81. Lines 64– 67 describes the unique role of nurses. In lines 269-279 we discuss intersectoral collboration in which a text that shows research findings that indicate that nurses tend to refer to doctors have now been added in lines 271-274
Comment: Missing from the manuscript is the information regarding the level of higher education at which curricular changes took place: bachelor or master? If bachelor as it can be assumed from Tables 1 and 2, what is taught at master level?
Response:
The authors have now provided a clarification in lines 103 – 105, and changed the title of Table 2 to show reflect undergraduate level.
Minor remarks:
Comments: Affiliations should be corrected as the names of institutes are capitalized in English.
Response: We are unsure what this comment means, however, the name of the institution is often written as the Swedish Red Cross University
Comment: What is „neglect of barn”? Table 2 Semester 5
Response: Corrected, should read child neglect.
Comment: English words should be used (Semester instead of Termin, Fig.1)
Response: Now reads Semester instead of Termin.
Comment: Fig. 1. does not require colors.
Response: Red filling now removed.
Comment: Table 2 contains Swedish text.
Response: The official translation of Swedish organisations have been used where possible; texts in Swedish have now been removed.

Reviewer 2 Report
Comments and Suggestions for Authors
First of all, I thank you for reviewing the study.
The study does not specify any research objectives.
In the methodology section: it is not specified what type of research has been carried out. A teaching model has been included in the Nursing Degree, but it is not known, for example, how it has been applied.
How were areas of the nursing programme identified where the content on violence against women and domestic violence fit naturally? The study refers to a content analysis of curriculum texts.
What sample was used for the study? Does this manuscript reflect an original research study? What are the characteristics of the sample? What are the inclusion and exclusion criteria?
What is the purpose for the analysis of the texts? How is this analysis analyzed by nursing students?
The discussion does not present similarities and/or differences with other studies. A presentation of results of different aspects is presented.
What are the limitations of the study?
What are the future lines of research?
The conclusions must meet the research objectives.
The manuscript must be adapted to a research study to be considered as such.
Author Response
Comment: The study does not specify any research objectives.
Response:
The focus of the paper is to describe the implementation process of the degree objective on VAW/DV in the nursing education at the Swedish Red Cross University and some of the lessons learned. Based the observations, a model for teaching the subject in nursing education is also proposed. A clarification has been provided with the insertion of words highlighted in yellow, see lines 98 – 101.
Comment: In the methodology section: it is not specified what type of research has been carried out. A teaching model has been included in the Nursing Degree, but it is not known, for example, how it has been applied.
Response:
- Now addressed, kindly see new text on lines 103- 105
- As mentioned in lines 220 - 224; 314 - 330; and 337 – 342, the model is simply to visualize our findings and proposed as a way of sharing our findings regarding the organization of VAW/DV teaching and learning activities at the university.
Comment:
How were areas of the nursing programme identified where the content on violence against women and domestic violence fit naturally? The study refers to a content analysis of curriculum texts.
Response:
- This is described in the methodology, specifically under “process”, see lines 133 – 139.
- Also, see the result section, specifically lines 192 – 213.
Comment:
What sample was used for the study? Does this manuscript reflect an original research study? What are the characteristics of the sample? What are the inclusion and exclusion criteria?
Response
This paper is based entirely on the analysis of documents, i.e., the undergraduate nursing curriculum and individual course syllabi from semesters one through to six of an undergraduate nursing program.
Comment:
What is the purpose for the analysis of the texts? How is this analysis analyzed by nursing students?
Response:
- See lines 118 – 131, more specifically, 127-131.
- Nursing students were not involved in the analysis, kindly see lines 140 – 146.
Comment The discussion does not present similarities and/or differences with other studies. A presentation of results of different aspects is presented.
Response: The current paper describes the process of implementing a new degree objective regarding violence against women and domestic violence in a nursing curriculum. The need to include VAW/ DV in nursing curriculum has long been established, however mandatory inclusion is still an emerging idea, thus there is a dearth of research that describes faculty experience of the implementation process. We have cited a few, see for example, lines 261 – 268.
Comment: What are the limitations of the study?
Response: Additional text, please see lines to 282- 287. Because the paper describes the implementation process of a relatively new additional national degree, the authors recognize this as uncharted waters. We have thus discussed the need for continued critical appraisal of the proposed model given some of challenges already reported in a national report (see lines, 315 - 330.
Comment: What are the future lines of research?
Response Similar to the comments above, kindly see lines 315 – 330.
Comment: The conclusions must meet the research objectives.
Response This requirement has already been met, kindly see additional text on lines 332 – 343.
Comment: The manuscript must be adapted to a research study to be considered as such.
Response The paper describes the implementation process of an additional degree objective for nursing and seven other educational programmes whose professions encounter victims and perpetrators of VAW/DV.

Reviewer 3 Report
Comments and Suggestions for Authors
Well written manuscript. Watts & Mayhew (2004) cited line 50: very pertinent but old reference. The paragraph about nurses' barriers to identifying VAW and DV does not include anything about the danger to the woman or partner if DV is revealed. Explanation of how this is incorporated into the content and evaluations might help to emphasize the danger to a woman or person in a DV living situation.
Expand on the descriptions of examinations in the narrative. The structural development of the model is described very well.
There are limitations included in the discussion. Adding a section for limitations would raise awareness of them and provide segue into considerations for future research.
Comments on the Quality of English LanguageTable 2 in Semester 4 column, Examination column includes a Swedish phrase.
Author Response
Reviewer 3
Comments and Suggestions for Authors
Comment: Well written manuscript. Watts & Mayhew (2004) cited line 50: very pertinent but old reference. The paragraph about nurses' barriers to identifying VAW and DV does not include anything about the danger to the woman or partner if DV is revealed. Explanation of how this is incorporated into the content and evaluations might help to emphasize the danger to a woman or person in a DV living situation.
Response There is now a text on the danger to the woman or partner if DV is revealed, please see lines 62 – 63. Yes we agree that Watts & Mayhew (2004) reference are old, but because they are pertinent like the Reviewer rightfully observed, we hope that we can retain this reference as we are yet to find a more recent reference that aptly describe the fact described.
Comment: Expand on the descriptions of examinations in the narrative. The structural development of the model is described very well.
Response To avoid repeating information on the table, a brief additional text is now available on lines 306 – 307
Comment: There are limitations included in the discussion. Adding a section for limitations would raise awareness of them and provide segue into considerations for future research.
Response Additional text to highlight limitations.
Comments on the Quality of English Language
Response Manuscript has been proofread again.
Comment: Table 2 in Semester 4 column, Examination column includes a Swedish phrase.
Response: Text has been removed.

Reviewer 4 Report
Comments and Suggestions for Authors
The submitted manuscript presents a very interesting topic. There is a real need for health professionals to have sufficient knowledge about violence.
As the authors say, curricula have been incorporating these issues for years.
The main problem I find in the manuscript is in the data analysis. The authors propose a content analysis. They explain well what are the steps to do this analysis. However, when the authors present the results, the information is not clear.
There is no theoretical framework of the researchers. This is necessary to guide the process.
Regarding coding, the sampling units, recording and context are not indicated.
Subsequently, categories do not emerge randomly. The categories are derived from the theoretical framework. Furthermore, they must be clearly defined (meaningful, exhaustive, mutually exclusive, clear and replicable).
Finally, the authors do not talk about reliability. They must ensure that different researchers, using the same data, arrive at a similar category coding system.
Author Response
Reviewer 4
Comments and Suggestions for Authors
The submitted manuscript presents a very interesting topic. There is a real need for health professionals to have sufficient knowledge about violence.
As the authors say, curricula have been incorporating these issues for years.
Response: The authors appreciate the constructive criticism and positive feedback.
Comment: The main problem I find in the manuscript is in the data analysis. The authors propose a content analysis. They explain well what are the steps to do this analysis. However, when the authors present the results, the information is not clear.
Response: Kindly see an updated Table one showing the process of content analysis. See also Table 2 which has now been updated to show how teaching is organized to reflect findings from Table 1.
Comment:
There is no theoretical framework of the researchers. This is necessary to guide the process.
Response
Considering the focus of this paper, Emancipatory nursing may be a relevant theory, however, the authors chose not to include a theory. This manuscript describes the process of implementation of a new degree objective in Sweden in 2017 that led to mandatory teaching of Violence against women and domestic violence (VAW/DV), and how it led to a model. Thus, the paper is more about describing an approach to developing a theoretical model for teaching VAW/DV rather than an attempt to prove an existing theory or framework.
Comment:
Regarding coding, the sampling units, recording and context are not indicated.
Response
Kindly see lines 103 - 105; 133 – 139; 148 - 153 and also an updated Table 1.
Comment
Subsequently, categories do not emerge randomly. The categories are derived from the theoretical framework. Furthermore, they must be clearly defined (meaningful, exhaustive, mutually exclusive, clear and replicable).
Response
The categories are not derived from any theoretical framework but rather from content analysis of curriculum and course syllabi of the undergraduate nursing programme. However, the teaching model – Dare to Ask! is derived from the authors’ findings from implementing the new degree objective at their institution.
Comments
Finally, the authors do not talk about reliability. They must ensure that different researchers, using the same data, arrive at a similar category coding system.
Response
Kindly see a clarification and additional texts on lines 308 – 313.

Round 2
Reviewer 1 Report
Comments and Suggestions for Authors
The authors have not changed the title of the paper in the revised manuscript. I very strongly urge them to do so and CHANGE IT as they consented in their answer. Currently it still states: Dare to Ask! A Model for TEACHING VIOLENCE Against Women....
The same title is used for Figure 1. This current title absolutely undermines the message that the authors try to convey.
I have read the authors' answers which gives me the impression that they have no wish to change the major goal of the paper (to show compliance with a national requirement rather than share their experience with colleagues regarding the enlargement and re-development of a curricular topic), I have only a few remarks.
One is that knowledge, attitudes and skills together establish competence, and there are countless examples in higher education that these can be and are separated for teaching purposes. So, one definitely can teach knowledge regarding VAW or DV without teaching skills or changing attitudes. It is a salient point because it would be perfectly reasonable to teach knowledge about VAW and DV for nurses at the bachelor level, and teach attitudes and skills at master level thereby establishing competence to deal with such cases for MSc nurses but not for BSc nurses.
From this, it follows that it would be very interesting to see how the new curriculum (in terms of at least the major topics) compares to the old curriculum.
A minor issue is that healthcare being one sector of the economy, synchronizing the teaching of a given topic between medical doctors and nursing students would count as intrasectoral (not intersectoral) collaboration.
Comments on the Quality of English LanguageAs I suggested in my previous review, the names of the institutes of the authors should be changed according to English grammar by capitalizing all parts of the name. That is, instead of "Department of health sciences", this should be written: "Department of Health Sciences", etc.
Author Response
Reviewer 1
Comment
The authors have not changed the title of the paper in the revised manuscript. I very strongly urge them to do so and CHANGE IT as they consented in their answer. Currently it still states: Dare to Ask! A Model for TEACHING VIOLENCE Against Women....The same title is used for Figure 1. This current title absolutely undermines the message that the authors try to convey.
Response
The title change now reflects in relevant areas.
Comments
I have read the authors' answers which gives me the impression that they have no wish to change the major goal of the paper (to show compliance with a national requirement rather than share their experience with colleagues regarding the enlargement and re-development of a curricular topic), I have only a few remarks.
Response
We must have missed any comments suggesting that the reviewer wanted the aim changed. The authors have now reworded the aim in line with the Reviewer’s suggestion 98 to 101.
Comment
One is that knowledge, attitudes and skills together establish competence, and there are countless examples in higher education that these can be and are separated for teaching purposes. So, one definitely can teach knowledge regarding VAW or DV without teaching skills or changing attitudes. It is a salient point because it would be perfectly reasonable to teach knowledge about VAW and DV for nurses at the bachelor level, and teach attitudes and skills at master level thereby establishing competence to deal with such cases for MSc nurses but not for BSc nurses.
Response
Attitudes are thought as part of basic facts about violence related to norms and values, not student’s attitudes. It is however important to improve student’s preparedness and willingness to screen and respond in line with the expectations of the degree objective. Kindly see our response to the important comment about BSc and MSc nursing competencies below.
Comment
One is that knowledge, attitudes and skills together establish competence, and there are countless examples in higher education that these can be and are separated for teaching purposes. So, one definitely can teach knowledge regarding VAW or DV without teaching skills or changing attitudes. It is a salient point because it would be perfectly reasonable to teach knowledge about VAW and DV for nurses at the bachelor level, and teach attitudes and skills at master level thereby establishing competence to deal with such cases for MSc nurses but not for BSc nurses.
Response
The comment about BSc and MSc is relevant, however, our job as educators is to implement the national directive as it is. Below is the expectation of undergraduate competence related to the national degree objective, Relevant professions included in the directive are expected to be able to identify cases and people at risk act, kindly see links to reports and organization below, (although the websites and documents are in Swedish, they can be translated for further reading). Do kindly see excerpt page 148 in the government document linked below,
The aim is for the students to gain knowledge of to prevent and detect violence, and also gain knowledge about interventions for those who expose others to violence and those who are exposed to violence.
https://github.jbcj.top:443/https/www.regeringen.se/contentassets/f837b6325e0c4f59a4d17cb5049bee58/en-nationell-strategi-for-att-forebygga-och-bekampa-mans-vald-mot-kvinnor_utdrag-ur-skr.-2016_17_10.pdf
(Swedish exercept: Målsättningen är att studenterna dels ska få kunskap om hur man kan förebygga och upptäcka våld, dels få kunskap om insatser för dem som utsätter andra för våld respektive dem som är utsatta för våld).
Furthermore, the National center, NCK, which is a government funded expert organization provides training videos and materials many of which are for BSc nurses. The reason, as described by the NCK in the link below is as follows:
“…In order to achieve good and adequate care, the cause of the symptoms for which the patient is seeking must be identified as far as possible. Few women spontaneously talk about their exposure to violence. The responsibility to identify exposure to violence as a contributing cause of the health problem therefore rests with the healthcare staff... ”.
(För att kunna uppnå god och adekvat vård måste så långt möjligt orsaken till de symtom som patienten söker för identifieras. Få kvinnor berättar spontant om sin våldsutsatthet. Ansvaret att identifiera våldsutsatthet som en bidragande orsak till hälsoproblemet ligger därför på vårdpersonalen).
https://github.jbcj.top:443/https/www.nck.uu.se/kunskapsbanken/webbstod-for-varden/halso-och-sjukvardens-ansvar/halso--och-sjukvardens-ansvar/
Moreover, knowledge lies at different levels, nurses are expected to identify cases and be able to respond via medical care whether in general practice or in specialized care, for example midwives and psychiatric nurses. Response within health can also be in the form of contacting social services, the police, and women shelters etc.,. Such non-medical response does not need specialization but clear routines to guide the health care team. Thus ability to recognize signs, asks the right follow up questions and be familiar with routines and guidelines for response is also a form of knowledge which, according to government documents, is important for BSc nurses to have. Reports regarding screening often include BSc nurse and specialist nurses.
https://github.jbcj.top:443/https/www.socialstyrelsen.se/globalassets/sharepoint-dokument/artikelkatalog/ovrigt/2018-3-28.pdf
Comment
A minor issue is that healthcare being one sector of the economy, synchronizing the teaching of a given topic between medical doctors and nursing students would count as intrasectoral (not intersectoral) collaboration.
Answer
True, thank you for the observation. Also, doctors and nurses are part of the care team with established areas of responsibility and hierarchies in terms of duties and decision-making process. Do kindly see lines 271 to 281 and other places where the reference to “intersectoral collaboration” is made, it is not nurses’ collaboration with doctors that is implied, but rather the importance of being aware of stakeholders outside healthcare. Although a description of the care team and its dynamics is not the focus of the current paper, additional texts have been added to lines 274 to 277 to highlight recommendations which reinforce the need for clear routines for response and decision making within the care team.
Comments on the Quality of English Language
As I suggested in my previous review, the names of the institutes of the authors should be changed according to English grammar by capitalizing all parts of the name. That is, instead of "Department of health sciences", this should be written: "Department of Health Sciences", etc.
Response
The updated manuscript now reflects the above recommendations.

Reviewer 2 Report
Comments and Suggestions for Authors
I still consider that this study does not fit into a quantitative, qualitative or mixed research methodology.
Author Response
Dear Reviewer,
Thank you for the following comments, i.e., "I still consider that this study does not fit into a quantitative, qualitative or mixed research methodology."
To respond we provide a few examples below where content analysis has been used to analyse curriculum and in some cases text books. Analysis of curriculum using content analysis is considered a qualitative research method.
Amini-Rarani, M., & Nosratabadi, M. (2021). Content analysis of the official curriculum of undergraduate degree in Iran's medical sciences universities from the perspective of social health: A qualitative study. Journal of Education and Health Promotion, 10(1).
Islam, K. M. M., & Asadullah, M. N. (2018). Gender stereotypes and education: A comparative content analysis of Malaysian, Indonesian, Pakistani and Bangladeshi school textbooks. PloS one, 13(1), e0190807.
Al-Shakarchi, N., Obolensky, L., Walpole, S., Hemingway, H., & Banerjee, A. (2019). Global health competencies in UK postgraduate medical training: a scoping review and curricular content analysis. BMJ open, 9(8), e027577.
In the data analysis section, we cite the references below as the rationale for this choice, also considered a form of qualitative research, do kindly see lines 149 - 152. We have merely only used a research that has been previously used within the research community.
46. Stemler S, Bebell D. An Empirical Approach to Understanding and Analyzing the Mission Statements of Selected Educational Institutions. 1999.
- Erlingsson C, Brysiewicz P. A hands-on guide to doing content analysis. African journal of emergency medicine. 2017;7(3):93-9.
Reviewer 4 Report
Comments and Suggestions for Authors
The manuscript has improved substantially with the modifications made by the authors.
Author Response
Dear Reviewer,
Many thanks for your useful comments.